# Comparison of the Efficacies of Direct-Acting Antiviral Treatment for HCV Infection in People Who Inject Drugs and Non-Drug Users

**DOI:** 10.3390/medicina58030436

**Published:** 2022-03-17

**Authors:** Jui-Ting Hsu, Ping-I Hsu, Chang-Bih Shie, Seng-Kee Chuah, I-Ting Wu, Wen-Wei Huang, Sheng-Yeh Tang, Kun-Feng Tsai, Li-Fu Kuo, Supratip Ghose, Jui-Che Hsu, Chih-An Shih

**Affiliations:** 1Division of Gastroenterology, Department of Internal Medicine, An Nan Hospital, China Medical University, Tainan 709, Taiwan; verybiggest80@gmail.com (J.-T.H.); williamhsup@yahoo.com.tw (P.-I.H.); cbshie@gmail.com (C.-B.S.); ilessalen@gmail.com (I.-T.W.); wh0531@yahoo.com.tw (W.-W.H.); d72460@mail.tmanh.org.tw (S.-Y.T.); d70714@mail.tmanh.org.tw (K.-F.T.); d71393@mail.tmanh.org.tw (L.-F.K.); 2Division of Hepato-Gastroenterology, Department of Internal Medicine, Kaohsiung Chang Gung Memorial Hospital, Kaohsiung 833, Taiwan; chuahsk@seed.net.tw; 3Department of Education and Research, An Nan Hospital, China Medical University, Tainan 709, Taiwan; sgresearch@gmail.com (S.G.); therealhsudandan@gmail.com (J.-C.H.); 4Division of Gastroenterology and Hepatology, Department of Internal Medicine, Antai Medical Care Corporation, Antai Tian-Sheng Memorial Hospital, Pingtung County 928, Taiwan; 5Department of Nursing, Meiho University, Pingtung County 912, Taiwan

**Keywords:** hepatitis C virus, people who inject drugs, direct-acting antiviral agents

## Abstract

*Background and Objectives*: Hepatitis C virus (HCV) is a major cause of liver disease worldwide. People who inject drugs (PWIDs) constitute the majority of patients with HCV infection in the United States and Central Asia. There are several obstacles to treating HCV infection in PWIDs because PWIDs are often accompanied by concurrent infection, low compliance, substance abuse, and risky behavior. The aim of the study is to compare the efficacies of direct-acting antiviral (DAA) therapy for HCV infection in PWIDs and those without opioid injection. *Materials and Methods*: In this retrospective cohort study, we included 53 PWIDs with HCV infections treated on site in a methadone program and 106 age- and sex-matched patients with HCV infections who had no history of opioid injection (ratio of 1:2). All eligible subjects received anti-HCV treatment by DAA agents in our hospital from March 2018 to December 2020. The charts of these patients were carefully reviewed for demographic data, types of DAA agents, and treatment outcomes. The primary outcome measure was sustained virological response (SVR). *Results*: PWIDs and non-drug users had different HCV genotype profiles (*p* = 0.013). The former had higher proportions of genotype 3 (18.9% vs. 7.5%) and genotype 6 (24.5% vs. 14.2%) than the latter. The two patient groups had comparable rates of complete drug refilling (100.0% vs. 91.1%) and frequency of loss to follow-up (3.8% vs. 0.9%). However, PWIDs had a lower SVR rate of DAA treatment than non-drug users (92.2% vs. 99.0%; *p* = 0.04). Further analysis showed that both human immunodeficiency virus (HIV) coinfection and history of PWID were risk factors associated with treatment failure. The subjects with coinfection with HIV had lower SVR rates than those without HIV infection (50.0% vs. 96.5%; *p* = 0.021). *Conclusions*: PWIDs with HCV infections have higher proportions of HCV genotype 3 and genotype 6 than non-drug users with infections. DAA therapy can achieve a high cure rate (>90%) for HCV infection in PWID, but its efficacy in PWID is lower than that in non-drug users.

## 1. Introduction

Hepatitis C virus (HCV) is a major cause of liver disease worldwide [1,2]. Globally, it is estimated that 71 million people have chronic HCV infections [3]. People who inject drugs (PWID) constitute the majority of patients with HCV infection in the United States [4] and in Central Asia [5]. The prevalence of HCV infection exceeds 60% in PWIDs [6,7,8]. A study from Taiwan also showed that 97.1% of 591 prisoners who injected drugs were anti-HCV antibody-positive [9].

Recently, HCV treatment has evolved from interferon-containing regimens to direct-acting antiviral (DAA) therapy [10,11,12]. The major advantages of current DAA therapy are minimal side effects and high sustained virological response (SVR) rates [13,14]. Some studies demonstrate a high efficacy of DAA treatment in PWIDs with SVR over 95% [15,16,17]. However, there are still many obstacles to the treatment of HCV infection in PWIDs because PWIDs are often accompanied with concurrent HIV infection, multiple-genotype infections, low compliance, substance abuse, risky behavior, and reinfection conditions [6]. Therefore, physicians are often concerned that these factors will affect the treatment outcome [18,19]. In a survey for the clinicians’ views of HCV treatment candidacy with DAA regimens for PWIDs [8], reinfection and medication cost were cited as the most important concerns when determining candidacy.

We hypothesized that PWIDs have lower adherence to DAA therapy and a lower HCV cure rate than patients who have no history of opioid injection and designed the study to compare the complete drug refilling rate and cure rate of DAA therapy for HCV infections in PWIDs and those without injection drug use.

## 2. Methods

### 2.1. Study Population

In this retrospective cohort study, we included 53 PWIDs with HCV infection treated on-site in a methadone program and 106 age- and sex-matched non-drug users with HCV infections (ratio of 1:2). In the PWID group, patients were included from methadone clinics in the An Nan Hospital of China Medical University. The eligible criteria included (1) adult patients (aged ≥ 18 years), (2) self-declaration of opioid injection within the past 90 days, (3) attendance in a methadone maintenance treatment program, (4) HCV infection documented by a positive result of serum HCV RNA testing, and (5) treatment of HCV infection by DAAs. In the non-PWID group, patients were from the gastroenterology clinics in the An Nan Hospital of China Medical University. The eligible criteria included (1) adult patients (aged ≥ 18 years), (2) no history of injection of opioid drugs in the past, (3) HCV infection documented by a positive result of serum HCV RNA testing, and (4) treatment of HCV infection by DAAs. All eligible subjects received anti-HCV treatment by DAA agents in the An Nan Hospital of China Medical University in Taiwan between March 2018 and December 2020. This study was conducted in accordance with the principles of good clinical practice from the Declaration of Helsinki. The study protocol was approved by the Institutional Review Board of the An Nan Hospital of China Medical University. The Institutional Review Board waived informed consent requirement of the study because it was a retrospective work.

### 2.2. Study Design

All of the patients received DAA therapy and clinical follow-up according to the standard guideline of the Taiwan Health Insurance Bureau. Blood samples were collected at baseline before DAA therapy, at the end of treatment, and 12 weeks after the end of DAA therapy. Samples were drawn from venous blood to determine the levels of HCV RNA, alanine aminotransferase (ALT), aspartate aminotransferase (AST), and platelet count. In addition, HCV antibody (anti-HCV), HCV genotype, and HBsAg were tested before DAA therapy. The biochemical data were measured on a multichannel autoanalyzer (Hitachi, Inc., Tokyo, Japan). Anti-HCV tests were measured using the ARCHITECT Anti-HCV assay (Abbott GmbH, Wiesbaden, Germany) with the ARCHITECT i2000SR. HBsAg was a serological marker for hepatitis B virus, detected by the ARCHITECT HBsAg Qualitative II Confirmatory assay (Abbott Ireland, Diagnostics Division, Sligo, Ireland) with the ARCHITECT i2000SR). Serum HCV RNA was detected using the Abbott RealTime HCV assay (Abbott Molecular Inc., Des Plaines, IL 60018, USA) with a lower limit of quantitation of 12 IU/mL. The Abbott Real-Time HCV Genotype II assay (GT II, Abbott Molecular, Des Plaines, IL, USA) was used to analyze the presence of HCV genotypes, and the a PLUS assay (Des Plains, IL, USA) was used when GT II results were ambiguous. The presence of HIV p24 antigen and HIV-1/2 antibodies were routinely tested in PWID by Abbott ARCHITECT HIV Ag/Ab Combo Assay (Abbott GmbH & Co. KG. Max-Planck-Ring 2, 65205, Wiesbaden, Germany) with the ARCHITECT i2000SR (Abbott Laboratories. 1915 Hurd Drive, Irving, TX, 75038 USA) in PWID. However, HIV testing was routinely performed in patients who have no history of injection drug use. 

The charts of these patients were carefully reviewed for demographic data including age, gender, HCV genotype, serum HCV viral load, liver biochemistry, hemogram, and regimen of DAA therapy. Additionally, the follow-up status and treatment response including serum HCV RNA levels at the end of DAA therapy and 12 weeks after treatment were recorded. Finally, the follow-up status and treatment response were compared between groups. Additionally, the host and virological factors related to the failure of SVR were analyzed.

### 2.3. Outcome Measurements

The primary outcome measurement was SVR defined as the absence of serum HCV RNA 12 weeks after the end of DAA treatment [20,21]. This end point was assessed in all the patients who received at least one dose of DAA agents. The secondary outcomes were the rate of complete drug refilling and rate of loss to follow-up, which might indirectly indicate the poor adherence of patients. Complete drug refilling was defined as refilling 100% of DAA pills. Loss to follow-up was defined as the absence of serum HCV RNA data at the 12th week after DAA treatment.

### 2.4. Statistical Analysis

The χ^2^ test or Fisher’s exact test was used to analyze categorical data. Student’s *t* test was used to analyze continuous variables and to give results as mean ± standard deviation (SD). A *p*-value less than 0.05 was considered statistically significant. Statistic Package for the Social Science (SPSS) (version 26.0 for Microsoft Windows) was used for all statistical analyses.

### 2.5. Ethics Statement

The study was approved by the Institutional Review Board of Tainan Municipal An-Nan Hospital (TMANH110-REC025) on 28 July 2021. Informed consent was waived.

## 3. Results

### 3.1. Study Population

From March 2018 to December 2020, 53 PWIDs with HCV infections and treated on site in a methadone program, and 106 non-drug users were included in this retrospective cohort study. Table 1 shows the demographic data of the two patient groups. The distribution of the HCV genotype was significantly different between the PWID group and non-drug use group (χ^2^ = 12.712; df = 4; *p* = 0.013). The PWID group had higher frequencies of genotype 3 (18.9% vs. 7.5%) and genotype 6 (24.5% vs. 14.2%) than the non-drug use group. In contrast, the latter had higher frequencies of genotype 1b (22.6% vs. 9.4%) and genotype 2 (39.6% vs. 24.5%). The two patient groups had no differences in other clinical characteristics including age, gender, presence of cirrhosis, HCV viral load, HB antigen status, liver biochemistries, prothrombin time, and platelet count. Table 2 lists the regimens of DAA therapy of the two patient groups. There were no differences in the types of DAA regimens between groups. Table 3 demonstrates the treatment outcomes of the PWID group and non-drug use group. The two patient groups showed no difference in the rate of complete drug refilling (100.0% vs. 91.1%; *p* = 1.000). Additionally, the PWID group and non-drug use group had comparable frequency of loss to follow-up (3.8% vs. 0.9%; *p* = 0.258). At the end of DAA treatment, the PWID group had a lower frequency of undetectable HCV RNA in serum than the non-drug use group (90.2% vs. 100.0%; *p* = 0.003). At the 12th week following DAA therapy, the former also had a lower SVR rate than the latter (92.2% vs. 99.0%, *p* = 0.040).

### 3.2. Host and Virological Factors Related to SVR

Table 4 summarizes the host and virological factors related to SVR rate in the DAA therapy for HCV infection. PWID and HIV coinfection were the risk factors associated with failure to achieve SVR. PWID had a lower SVR rate than non-drug users (92.2% vs. 99.0%; *p* = 0.040). In this study, 8 out of 106 patients in the non-PWID group received HIV testing. All eigth patients had a negative result of HIV infection. In PWID group, 52 out of 53 patients received examination for HIV infection. The frequency of HIV coinfection in the PWID group was 7.7% (4/52). Overall, there were 60 patients in this study receiving HIV testing. Because 2 of the 60 patients with HIV testing did not receive complete follow-up, only 58 patients were included in the analysis for the impact of HIV-coinfection on the outcome of SVR. The subjects coinfected with HIV had a lower SVR rate than those without HIV infection (50.0% vs. 96.3%; *p* = 0.021). Other parameters such as age, sex, HCV genotype, initial viral load, initial ALT level, and DAA regimens were not related to SVR.

## 4. Discussion

In this study, we conducted a retrospective cohort study to investigate the efficacies of DAA therapy in PWIDs and non-drug users. The results demonstrated several important findings. First, PWIDs with HCV infections and non-drug users had different HCV genotype profiles. Second, PWIDs and those without drug injection had comparable rates of complete DAA refilling and frequency of loss to follow-up. Third, PWIDs had lower SVR rates than non-drug users in the DAA treatment for HCV infection. 

PWIDs have a high prevalence of HCV infection [6,7,8]. In the current study, the profiles of HCV genotypes were quite different between PWIDs and non-drug users. The PWID group had higher frequencies of genotype 3 (18.9% vs. 7.5%) and genotype 6 (24.5% vs. 14.2%) than the non-drug use group. The data were consistent with previous studies, which showed that genotype 3 was a popular strain in PWIDs with HCV infections [22]. Additionally, Jang et al. also reported that PWIDs with HCV infections had a higher frequency of genotype 6 than people with infections without intravenous drug use in Taiwan [23]. The aforementioned data suggest that choosing a DAA regimen covering genotypes 3 and 6 is important in the treatment of HCV infection in PWIDs.

PWIDs constitute the majority of people with HCV infections in the United States [4]. However, access to DAAs remains limited for PWIDs. The majority of PWIDs have not been treated often due to concerns about low adherence, substance abuse, risky behavior, and reinfection with HCV [6]. A previous study showed that drug adherence of DAA therapy in PWIDs was suboptimal (78%) [24]. In this real-world retrospective study, drug adherence of PWIDs could be assessed exactly. However, the complete drug refilling rates of the PWID treated on-site in a methadone program and non-drug users were 100% and 91.1%, respectively. The PWID group and non-drug use group did not have differences in the rate of complete drug refilling. Additionally, the frequencies of loss to follow-up in the PWID group and non-drug use group were 3.8% and 0.9%, respectively. The two patient groups also had no differences in the frequency of loss to follow up. 

Norton et al. showed that the SVR rate of DAA therapy for HCV infection was 96% in PWID though drug adherence was only 78% [24]. In the current study, DAA therapy also achieved a high cure rate (>90%) for HCV infection. However, the SVR rate in the PWID group was lower than that in the non-drug use group (92.2% vs. 99.0%). This study is the first demonstrating that PWIDs have a lower SVR rate than non-drug users in the DAA treatment for HCV infection. Nonetheless, our data still encourage clinicians to treat HCV infection because the SVR rate of DAA therapy was more than 90% in PWIDs. 

In this study, HIV coinfection was associated with failure to achieve SVR by DAA therapy. The SVR rates in the subjects with and without HIV coinfection were 50.0% and 96.3%, respectively. In another real-world setting, Gayam et al. also reported that DAA regimens had lower SVR at week 12 in HCV/HIV co-infection than in HCV monoinfection [25]. However, several previous studies showed that sofosbuvir-based DAA therapy for HCV infection in patients with HIV coinfection was highly effective and safe [26,27,28,29]. Possible factors of reduced response of DAA therapy in patients with HCV/HIV coinfection include poor drug adherence, HCV reinfection, immunodeficiency, and drug–drug interaction between DAAs and antiretroviral therapy (ART) [30,31,32,33]. In this study, two out of four patients in the PWID group with coinfections of HCV and HIV failed to cure their HCV infection by DAAs. One of them received glecaprevir/pibrentasvir (GLE/PIB) treatment for HCV infection, and abacavir, didanosine, and efavirenz treatment for HIV infection. The other received sofosbuvir/velpatasvir (SOF/VEL) treatment for HCV infection, and abacavir, lamivudine, and raltegravir treatment for HIV infection. Because nucleoside reverse transcriptase inhibitors (NNRTIs) such as efavirenz, etravirine, and nevirapine may reduce serum levels of GLE/PIB [34,35,36], the drug–drug interaction between efavirenz and GLE/PIB was a possible factor related to treatment failure in the patient with HIV/HCV coinfection treated with GLE/PIB, abacavir, didanosine, and efavirenz. Further studies are needed to clarify whether drug–drug interaction, HIV infection alone, or poor drug adherence in the subjects with HIV infections are the real causes of reduced response of DAA treatment for HCV infection. 

This study has some limitations. First, the current work was not a randomized controlled trial in which selection bias could be minimized. Second, the number of patients with HCV treatment failure was too small to perform a multivariate analysis to search for independent risk factors predicting eradication failure. Third, the drug adherence of patients taking DAAs could not be assessed exactly because this study was a retrospective study. Fourth, the stage of liver fibrosis is a possible factor influencing the outcome of anti-HCV therapy. In this study, only three patients in the PWID group and five patients in the non-drug use group received Fibroscan to assess the stage of liver fibrosis. None of the patients in the PWID group or non-drug use group receive liver biopsy to evaluate liver fibrosis. Since the data concerning liver fibrosis staging were not available in most of the patients, it was difficult to include this parameter in the analysis for risk factors influencing the outcome of sustained virological response. Nonetheless, the current study is the first work comparing the efficacies of DAA therapy for HCV infection in PWIDs and non-drug users.

In conclusion, PWIDs with HCV infections have higher proportions of HCV genotype 3 and genotype 6 than infected non-drug users. Although the efficacy of HCV treatment in PWID is lower than that in non-drug users, DAA therapy can still provide of a high cure rate (>90%) for HCV infection in PWID. The findings in this study encourage clinicians to treat HCV infection of PWID in the DAA era. In this study, HIV coinfection is a risk factor associated with failure of treatment by DAA therapy. It merits further investigation to clarify whether improvement of drug adherence or avoiding drug–drug interaction of DAA therapy can increase the cure rate of HCV infection in patients with HIV.

## Figures and Tables

**Table 1 medicina-58-00436-t001:** Demographic data of the PWID and non-drug use groups.

Characteristic	PWID Group (*n* = 53)	Non-Drug Use Group (*n* = 106)	χ^2^ (df)	*p*-Value
Age—yr			t = 0.216	0.829
Mean	50.4	50.1		
Range	32–68	36–68		
Male sex—no. (%)	48/53 (90.6%)	96/106 (90.6%)	0.000 (1)	1.000
HCV genotype no. (%)			12.712 (4)	0.013 *
1a	12 (22.6%)	17 (16.0%)		
1b	5 (9.4%)	24 (22.6%)		
2	13 (24.5%)	42 (39.6%)		
3	10 (18.9%)	8 (7.5%)		
4	0	0		
5	0	0		
6	13 (24.5%)	15 (14.2%)		
HCV RNA—log_10_ IU/mL	6.10 ± 1.08	5.95 ± 1.10	t = 0.833	0.406
Cirrhosis—no. (%)	10 (18.9%)	25 (23.6%)	0.458 (1)	0.499
Liver biochemistry				
AST	63.6 ± 47.0	53.00 ± 34.1	t = 1.617	0.108
ALT	66.3 ± 44.4	73.9 ± 59.2	t = −0.899	0.414
Albumin	4.2 ± 0.3	4.6 ± 3.2	t = −0.912	0.363
Total bilirubin	0.68 ± 0.46	0.72 ± 0.30	t = −0.700	0.485
Prothrombin time (INR)	1.03 ± 0.08	1.12 ± 0.88	t = −0.725	0.469
Platelet count < 90,000 per mm^3^ (%)	2 (3.8%)	5 (4.7%)	-	1.000
HBV carrier—no. (%)	5 (9.4%)	13 (12.3%)	0.282 (1)	0.595
HIV coinfection	4/52 (7.7%)	0/8 (0%)	-	1.000

* Denotes significant difference.

**Table 2 medicina-58-00436-t002:** DAA regimens for HCV infection in the PWID and non-drug user groups.

Anti-HCV DAA Therapy	PWID Group (*n* = 53)	Non-Drug Use Group (*n* = 106)	*p*-Value
Regimens			0.204
Ledipasvir + Sofosbuvir	9 (17.0%)	23 (21.7%)	
Glecaprevir + Pibrentasvir	22 (41.5%)	39 (36.9%)	
Sofosbuvir + Velpatasvir	16 (30.2%)	40 (37.7%)	
Elbasvir + Grazoprevir	0 (0%)	2 (1.9%)	
Paritaprevir + Ritonavir + Ombitasvir + Dasabuvir	1 (1.9%)	0 (0%)	
Ledipasvir + Sofosbuvir +Ribavirin	2 (3.8%)	1 (0.9%)	
Elbasvir + Grazoprevir + Ribavirin	0 (0%)	1 (0.9%)	
Sofosbuvir + Velpatasvir + Ribavirin	2 (3.8%)	0 (0%)	
Sofosbuvir + Ribavirin	1 (1.9%)	0 (0%)	

**Table 3 medicina-58-00436-t003:** Treatment outcomes of DAA therapy in PWID and non-drug user groups.

Outcomes	PWID Group (*n* = 53)	Non-Drug Use Group (*n* = 106)	*p*-Value
Undetectable HCV RNA at the end of treatment—no. (%)	46/51 (90.2%)	106/106 (100.0%)	0.003 *
Sustained virological response—no. (%)	47/51 (92.2%)	104/105 (99.0%)	0.040 *
Complete drug refilling—no. (%)	53/53 (100.0%)	105/106 (91.1%)	1.000
Loss to follow up—no. (%)	2/53 (3.8%)	1/106 (0.9%)	0.258

* Denotes significant difference.

**Table 4 medicina-58-00436-t004:** Clinical and virological factors related to sustained virological response.

Variable	Number of Patients	Sustained Virological Response No. (%)	*p*-Value
Age			1.000
<60 yr	139	134 (96.4%)	
≧60 yr	17	17 (100%)	
Sex			0.401
Male	141	137 (97.2%)	
Female	15	14 (93.3%)	
PWID			0.040 *
(−)	105	104 (99.0%)	
(+)	51	47 (92.2%)	
HCV genotype no. (%)			0.852
1a	27	26 (96.3%)	
1b	29	29 (100%)	
2	55	53 (96.4%)	
3	18	17 (94.4%)	
4	0	0	
5	0	0	
6	27	26 (96.3%)	
HCV RNA			0.657
<800,000 IU/mL	55	54 (98.2%)	
≥800,000 IU/mL	101	97 (96.0%)	
ALT			1.000
≤1X ULN	85	82 (96.5%)	
>1X ULN	71	69 (97.2%)	
HIV coinfection ^†^			0.021 *
No	54	52 (96.3%)	
Yes	4	2 (50.0%)	
DAA treatment			0.986
Ledipasvir + Sofosbuvir	32	32 (100.0%)	
Glecaprevir + Pibrentasvir	59	57 (96.6%)	
Sofosbuvir + Velpatasvir	55	52 (94.5%)	
Others	10	10 (100.0%)	

* Denotes significant difference; ^†^ only 58 patients were included for the analysis because two out of the 60 patients receiving HIV testing did not receive complete follow-up.

## Data Availability

Raw data supporting the findings of this study are available from the corresponding author on request and Appendix A.

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
