# Peer review of "Comparison of the Efficacies of Direct-Acting Antiviral Treatment for HCV Infection in People Who Inject Drugs and Non-Drug Users"

_medicina, 2022, doi:10.3390/medicina58030436_

Round 1

Reviewer 1 Report

Have you assessed the liver fibrosis staging, either by liver biopsy or by noninvasive methods? If so, please add the results to the analysis - the study group characteristics and as a factor potentially influencing the treatment efficacy.

Table 4 - what about the frequency of HIV infection in non-PWID? Other factors were analysed in the entire group.

Please, add the information about the schedules of antiretroviral therapy/DAA in HIV-infected patients without SVR - if there were any drug-drug interactions.

117-118 - should be 'virological' instead of 'bacterial' factors

Author Response

REPLY TO THE COMMENTS

* Reply to the reviewer 1

Comment 1: Have you assessed the liver fibrosis staging, either by liver biopsy or by noninvasive methods? If so, please add the results to the analysis - the study group characteristics and as a factor potentially influencing the treatment efficacy.
Reply to comment 1: In this study, only 3 patients in the PWID group and 5 patients in the non-drug use group received Fibroscan to assess the stage of liver fibrosis. None of the patients in the PWID group or non-drug use group receive liver biopsy. Since the data concerning liver fibrosis staging were not available in most of the patients, it was difficult to include this parameter in the analysis for risk factors influencing the outcome of sustained virological response. We have mentioned this point in the limitation section (P15, line 13: Fourth, stage of liver fibrosis is a possible factor influencing the outcome of anti-HCV therapy. In this study, only 3 patients in the PWID group and 5 patients in the non-drug use group received Fibroscan to assess the stage of liver fibrosis. None of the patients in the PWID group or non-drug use group receive liver biopsy to evaluate liver fibrosis. Since the data concerning liver fibrosis staging were not available in most of the patients, it was difficult to include this parameter in the analysis for risk factors influencing the outcome of sustained virological response.), and sincerely thank the reviewer’s valuable comments! 

Comment 2: Table 4 - what about the frequency of HIV infection in non-PWID? Other factors were analysed in the entire group.

Reply to comment 2: In non-PWID group, 8 out of 106 patients received HIV testing. All the 8 patients receiving HIV testing have negative results (Please see Table 1). In PWID group, 52 out of 53 patients received examination for HIV infection. The frequency of HIV co-infection in PWID group was 7.7% (4/52). Overall, there were 60 patients in this study receiving HIV testing. Because 2 of the 60 patients with HIV testing didn’t receive complete follow-up, only 58 patients were included in the analysis for the impact of HIV-coinfection on the outcome of sustained virological response. We have mentioned this point in the revised manuscript (P11, line 12: In this study, 8 out of 106 patients in the non-PWID group received HIV testing. All the 8 patients had a negative result of HIV infection. In PWID group, 52 out of 53 patients received examination for HIV infection. The frequency of HIV coinfection in PWID group was 7.7% (4/52). Overall, there were 60 patients in this study receiving HIV testing. Because 2 of the 60 patients with HIV testing didn’t receive complete follow-up, only 58 patients were included in the analysis for the impact of HIV-coinfection on the outcome of SVR.), and Table 4 (P28, line 5: † Only 58 patients were included for the analysis because two out of the 60 patients receiving HIV testing didn’t receive complete follow-up.).

Comment 3. Please, add the information about the schedules of antiretroviral therapy/DAA in HIV-infected patients without SVR - if there were any drug-drug interactions.

Reply to comment 3: In this study, two out of 4 patients in PWID group having coinfection of HCV and HIV failed to cure HCV infection by DAAs. One of them received glecaprevir/pibrentasvir (GLE/PIB) treatment for HCV infection, and abacavir, didanosine and efavirenz treatment for HIV infection. The other received sofosbuvir/velpatasvir (SOF/VEL) treatment for HCV infection, and abacavir, didanosine and raltegravir treatment for HIV infection. Because nucleoside reverse transcriptase inhibitors (NNRTIs) such as efavirenz, etravirine and nevirapine may reduce serum levels of GLE/PIB (World J Clin Cases 2021;9:4491-99; J Infect Dis 2020;221:223-31), the drug-drug interaction between efavirenz and GLE/PIB was a possible factor related to treatment failure in the patient with HIV/HCV coinfection treated with GLE/PIB, abacavir, didanosine and efavirenz. We have addressed the issue of drug-drug interaction in the revised manuscript (P14, line 6 from the bottum:

In this study, two out of 4 patients in PWID group having coinfection of HCV and HIV failed to cure HCV infection by DAAs. One of them received glecaprevir/pibrentasvir (GLE/PIB) treatment for HCV infection, and abacavir, didanosine and efavirenz treatment for HIV infection. The other received sofosbuvir/velpatasvir (SOF/VEL) treatment for HCV infection, and abacavir, lamivudine and raltegravir treatment for HIV infection. Because nucleoside reverse transcriptase inhibitors (NNRTIs) such as efavirenz, etravirine and nevirapine may reduce serum levels of GLE/PIB34-36, the drug-drug interaction between efavirenz and GLE/PIB was a possible factor related to treatment failure in the patient with HIV/HCV coinfection treated with GLE/PIB, abacavir, didanosine and efavirenz.).  

Comment 4. 117-118 - should be 'virological' instead of 'bacterial' factors.

Reply to comment 4: According the reviewer’s comments, we have revised the sentences concerning virological factors (P11, line 8: Host and virological factors related to SVR;P11, line 9: Table 4 summarizes the host and virological factors related to SVR rate).

We sincerely thank the reviewer’s valuable and constructive comments!

Reviewer 2 Report

Dear Authors,

The work entitled “Comparison of the Efficacies of Direct-Acting Antiviral Treatment for HCV Infection in People Who Inject Drugs and Non-Drug Users” has has merit, but some improvements should be made to make it suitable for publication.

General comments:

The study is relevant, but the methodology needs careful revision, adding the step-by-step of the experiments performed (or detailing what was obtained through secondary data).

English needs to be revised.

Abstract

Lines 19-20: The objectives could be better written. Ex: “The aim of this study is...”

What do the authors mean by "...those without drug injection”? Please, clarify the objetives in the abstract.

Introduction

Lines 51-52: “Therefore, physicians often concern that these factors will affect the treatment efficacy, and hesitate to provide DAA treatment for HCV-infected PWID” please provide reference for this sentence. Physicians' hesitancy to provide adequate treatment to a patient is not acceptable, regardless of whether the patient is a drug user or has some risky behavior. Please clarify this point.

Lines 53-55: The objectives should be clearly defined. Please provide more detail on the hypothesis and goals of the study.

Methods

-Study population: What criteria were adopted for classification as PWID? Self-declaration? Please, clarify this point in the text.

Lines 60-61: “All the eligible subjects received anti-HCV treatment by DAA agents in our hospital”. What do the authors mean as “our hospital”? Please provide more information about the patient recruitment site.

-Ethics statement: Was the study performed according to Helsinki Declaration guidelines?

Why was informed consent waived?

- Methods section lacks fundamental points of the study such as HCV serological and molecular analyzes. Ex: How was the HCV quantification and the HCV genotyping performed? All serological and molecular analyzes must be detailed in ‘methods’. In addition, it is not explained how the biochemical, histological and coinfection (as HBV and HIV coinfections) data were obtained.

Results:

-Suggestion: A phylogenetic analysis to elucidate whether HCV genotypes 3 and 6 found in this study are genetically related to HCV  isolates  from PWID from other studies or to other vulnerable populations would add value to the manuscript.

Conclusions:

- The conclusions summarize the results, but do not provide new insights into the importance of HCV in PWID.

Author Response

REPLY TO THE COMMENTS

* Reply to the reviewer 2

General comments:

Comment 1: The study is relevant, but the methodology needs careful revision, adding the step-by-step of the experiments performed (or detailing what was obtained through secondary data).

Reply to comment 1: We sincerely thank the reviewer’s valuable comments and have carefully revised the manuscript and added the step-by-step of the experiments performed (e.g., P6, line 9: In the PWID group, patients were included from methadone clinics in the An Nan Hospital of China Medical University. The eligible criteria included (1) adult patients (aged ³ 18 years), (2) self-declaration of opioid injection within the past 90 days, (3) attendance in methadone maintenance treatment program, (4) HCV infection documented by a positive result of serum HCV RNA testing, and (5) treatment of HCV infection by DAAs. In the non-PWID group, patients were from the Gastroenterology clinics in the An Nan Hospital of China Medical University. The eligible criteria included (1) adult patients (aged ³ 18 years), (2) no history of injection of opioid drugs in the past, (3) HCV infection documented by a positive result of serum HCV RNA testing, and (4) treatment of HCV infection by DAAs.; P7, line 12: Blood samples were collected at baseline before DAA therapy, at the end of treatment and 12 weeks after the end of DAA therapy. Samples were drawn from venous blood to determine the levels of HCV RNA, alanine aminotransferase (ALT), aspartate aminotransferase (AST), and platelet count. In addition, HCV antibody (anti-HCV), HCV genotype, and HBsAg were tested before DAA therapy.).

.

Comment 2. English needs to be revised.

Reply to comment 2: According to the reviewer’s valuable comments, the English of the manuscript has been revised by our colleague who is a native English speaker, and all the grammatical and writing style errors in the original version have been corrected (e.g., P5, line 13: PWID are often accompanied with concurrent HIV infection.; P9, line 4: The primary outcome measurement; P11, line 2: Additionally, the PWID group and non-drug use group had comparable frequency of loss to follow-up).  

Abstract

Comment 1. Lines 19-20: The objectives could be better written. Ex: “The aim of this study is...”

Reply to comment 1: We have re-written the objectives according to the comments of the reviewer (P3, line 9: The aim of the study is to compare the efficacies of direct-acting antiviral (DAA) therapy for HCV infection in PWID and those without opioid injection.).

Comment 2. What do the authors mean by "...those without drug injection”? Please, clarify the objetives in the abstract.

Reply to comment 1: We have clarified the objectives in the control group (P 3, line 12: 106 age- and sex-matched HCV-infected patients who had no history of opioid injection).

Introduction

Comment 1. Lines 51-52: “Therefore, physicians often concern that these factors will affect the treatment efficacy, and hesitate to provide DAA treatment for HCV-infected PWID” please provide reference for this sentence. Physicians' hesitancy to provide adequate treatment to a patient is not acceptable, regardless of whether the patient is a drug user or has some risky behavior. Please clarify this point.

Reply to comment 1: (1) We have provided references (P20, line 4 from the bottom: Reference 18: Treloar C, Newland J, Rance J, et al. Uptake and delivery of hepatitis C treatment in opiate substitution treatment: perceptions of clients and health professionals. J Viral Hepat 2010; 17:839–44; Reference 19: Asher AK, Portillo CJ, Cooper BA, et al. Clinicians' views of hepatitis C virus treatment candidacy with direct-acting antiviral regimens for People Who Inject Drugs. Subst Use Misuse 2016;51:1218-23) for the sentence of “Therefore, physicians often concern that these factors will affect the treatment outcome.”.  (2) We agree with the reviewer’s valuable comments and have revised the sentence as “In a survey for the clinicians’ views of HCV treatment candidacy with DAA regimens for PWID,19 reinfection and medication cost were cited as most important concerns when determining candidacy.”(P5, line 3 from the bottom).

Comment 2. Lines 53-55: The objectives should be clearly defined. Please provide more detail on the hypothesis and goals of the study.

Reply to comment 2: According the valuable comments, we have provided the hypothesis and goals of the study in the revised manuscript (P6, line 1: We hypothesized that PWID have lower adherence to DAA therapy and a lower HCV cure rate than the patients who have no history of opioid injection, and designed the study to compare the complete drug refilling rate and cure rate of DAA therapy for HCV infection in PWID and those without injection drug use.).

Methods

Comment 1. Study population: What criteria were adopted for classification as PWID? Self-declaration? Please, clarify this point in the text.

Reply to comment 1: According to the comments, we have clarified the criteria adopted for classification as PWID in the revised manuscript (P6, line 9: In the PWID group, patients were from a methadone clinic in the An Nan Hospital of China Medical University. The eligible criteria included (1) adult patients (aged ³ 18 years), (2) self-declaration of injection of opioid drugs within the past 90 days, (3) attendance in methadone maintenance treatment program in the An Nan Hospital of China Medical University, (4) HCV infection documented by a positive result of serum HCV RNA testing, and (5) treatment of HCV infection by DAAs).  Additionally, we also described the criteria adopted for classification as non-PWID (P6, line 4 from the bottom: In the non-PWID group, patients were from the Gastroenterology clinics in the An Nan Hospital of China Medical University. The eligible criteria included (1) adult patients (aged ³ 20 years), (2) no history of injection of opioid drugs in the past, (3) HCV infection documented by a positive result of serum HCV RNA testing, and (4) treatment of HCV infection by DAAs.).

Comment 2. Lines 60-61: “All the eligible subjects received anti-HCV treatment by DAA agents in our hospital”. What do the authors mean as “our hospital”? Please provide more information about the patient recruitment site.

Reply to comment 2: In this study, PWID were included from a methadone clinic in the An Nan Hospital of China Medical University. In the non-PWID group, patients were from the Gastroenterology clinics in the An Nan Hospital of China Medical University. According to the reviewer’s comments, we have clearly described the recruitment site in the revised manuscript (P6, line 9: In the PWID group, patients were included from methadone clinics in the An Nan Hospital of China Medical University.; P6, line 4 from the bottom: In the non-PWID group, patients were from the Gastroenterology clinics in the An Nan Hospital of China Medical University.; P7, line 1: All the eligible subjects in the PWID and non-PWID groups received anti-HCV treatment by DAA agents in the An Nan Hospital of China Medical University in Taiwan between March 2018 and December 2020.).  

Comment 3. Ethics statement: Was the study performed according to Helsinki Declaration guidelines?

Reply to comment 3: Yes, the study was performed in accordance with the principles of good clinical practice from the Declaration of Helsinki. We have added the ethics statement in the revised manuscript (P7, line 4: This study was conducted in accordance with the principles of good clinical practice from the Declaration of Helsinki. The study protocol was approved by the Institutional Review Board of the An Nan Hospital of China Medical University.).

Comment 4. Why was informed consent waived?

Reply to comment 4: The Institutional Review Board waived informed consent requirement of the study because it was a retrospective work. We have addressed this point in the revised manuscript (P7, line 6: The Institutional Review Board waived informed consent requirement of the study because it was a retrospective work.).

Comment 5. Methods section lacks fundamental points of the study such as HCV serological and molecular analyzes. Ex: How was the HCV quantification and the HCV genotyping performed? All serological and molecular analyzes must be detailed in ‘methods’. In addition, it is not explained how the biochemical, histological and coinfection (as HBV and HIV coinfections) data were obtained.

Reply to comment 5: According to the reviewer’s important comments, we have added the fundamental points of the study such as HCV serological and molecular analyzes. In addition, we also explained how the biochemical, histological and coinfection (as HBV and HIV coinfections) data were obtained (P7, line 12: Blood samples were collected at baseline before DAA therapy, at the end of treatment and 12 weeks after the end of DAA therapy. Samples were drawn from venous blood to determine the levels of HCV RNA, alanine aminotransferase (ALT), aspartate aminotransferase (AST), and platelet count. In addition, HCV antibody (anti-HCV), HCV genotype, and HBsAg were tested before DAA therapy, and an abdominal sonography was performed to assess the presence of cirrhosis. The biochemical data were measured on a multichannel autoanalyzer (Hitachi, Inc., Tokyo, Japan). HCV antibodies (anti-HCV) were measured by a third-generation enzyme immunoassay (Abbott Labora-tories, North Chicago, Illinois). HBsAg was detected by commercial enzyme immunoassay (HBsAg; Abbott, Chicago, IL, USA). HCV RNA levels were determined with commercially available real-time PCR assays (RealTime HCV, low limit of detection, 12 IU/mL).  The Abbott Real-Time HCV Genotype II assay (GT II, Abbott Molecular, Des Plaines, Illinois) was used to analyze the presence of HCV genotypes, and the a PLUS assay (Des Plains, Illinois) was used when GT II results were ambiguous. The presence of HIV p24 antigen and HIV-1/2 antibodies were routinely tested in PWID by Abbott ARCHITECT HIV Ag/Ab Combo Assay (Abbott GmbH & Co. KG. Max-Planck-Ring 2, 65205, Wiesbaden, Germany) with the ARCHITECT i2000SR (Abbott Laboratories. 1915 Hurd Drive, Irving TX, 75038 U.S.A.) in PWID. However, HIV testing was routinely performed in patients who have no history of injection drug use.). 

Results:

Comment 1: Suggestion: A phylogenetic analysis to elucidate whether HCV genotypes 3 and 6 found in this study are genetically related to HCV isolates from PWID from other studies or to other vulnerable populations would add value to the manuscript.

Reply to comment 1: We sincerely thank the reviewer’s valuable comments that a phylogenetic analysis to elucidate whether HCV genotypes 3 and 6 found in this study are genetically related to HCV isolates from PWID from other studies or to other vulnerable populations can provide valuable information concerning overlapping social behaviors among vulnerable population groups. Because the important work needs specialists in this field and the sequencing data of HCV in other studies, we have not conduct the analysis in this study, and would like to perform such studies with other specialist in the future.

Conclusions

Comment 1. The conclusions summarize the results, but do not provide new insights into the importance of HCV in PWID.

Reply to comment 1: According to the important comments of reviewer, we have provide new insights of this study in the Conclusion section (P16, line 4: Although the efficacy of HCV treatment in PWID is lower than that in non-drug users, DAA therapy still can provide of a high cure rate (> 90%) for HCV infection in PWID. The findings in this study encourage clinicians to treat HCV infection of PWID in the DAA era. In this study, HIV coinfection is a risk factor associated with failure of treatment by DAA therapy. It merits further investigation to clarify whether improvement of drug adherence or avoiding drug-drug interaction of DAA therapy can increase cure rate of HCV infection in HIV patients.).  

We sincerely thank the reviewer’s valuable and constructive comments!

Round 2

Reviewer 2 Report

Dear Authors,

The manuscript has improved considerably, however, some changes would be important to make it suitable for publication.

-English minor revision is still needed.

- Methodology is still incomplete. The only kit whose name was provided the “Abbott Real-Time HCV Genotype II assay”. Which commercial immunoassay was used for the evaluation of anti-HCV and HBsAg? It is also essential to mention that HBsAg is a serological marker for hepatitis B virus.

- “HCV RNA levels were determined with commercially available real-time PCR assays (RealTime HCV, low limit of detection, 12 IU/mL)”. Which kit was used? Please, provide de description.

Author Response

REPLY TO THE COMMENTS

* Reply to the reviewer 2

General comments:

Comment 1: English minor revision is still needed.

Reply to comment 1: According to the reviewer’s valuable comments, the English of the manuscript has been revised by our colleague again who is a native English speaker, and all the grammatical and writing style errors in the original version have been corrected (e.g., P3, line 6: in the United States and Central Asia.; P5. line 12: the treatment of HCV infection in PWID; P5, line 4 from the bottom: Therefore, physicians are often concerned that these factors will affect treatment outcome.; P14, line 10: However, several previous studies showed that sofosbuvir-based DAA therapy for HCV infection in patients with HIV coinfection was highly effective and safe.).  

Comment 2. Methodology is still incomplete. The only kit whose name was provided the “Abbott Real-Time HCV Genotype II assay”. Which commercial immunoassay was used for the evaluation of anti-HCV and HBsAg? It is also essential to mention that HBsAg is a serological marker for hepatitis B virus.

Reply to comment 2: The commercial immunoassay of anti-HCV and HBsAg were used with ARCHITECT Anti-HCV assay and ARCHITECT HBsAg Qualitative II Confirmatory assay in our study, respectively. (P7, line 1 from the bottom: Anti-HCV tests were measured using the ARCHITECT Anti-HCV assay (Abbott GmbH, Wiesbaden, Germany) with the ARCHITECT i2000SR.; P8, line 2: HBsAg is a serological marker for hepatitis B virus, detected by the ARCHITECT HBsAg Qualitative II Confirmatory assay (Abbott Ireland, Diagnostics Division, Sligo, Ireland) with the ARCHITECT i2000SR.).

Comment 3. “HCV RNA levels were determined with commercially available real-time PCR assays (RealTime HCV, low limit of detection, 12 IU/mL)”. Which kit was used? Please, provide de description.

Reply to comment 3: The Abbott RealTime HCV assay was used to detect the HCV viral load in our study (P8, line 5: Serum HCV RNA was detected using the Abbott RealTime HCV assay (Abbott Molecular Inc. Des Plaines, IL 60018 USA) with a lower limit of quantitation of 12 IU/mL).

We sincerely thank the reviewer’s valuable and constructive comments!
